

# The impact of mental and somatic stressors on physical activity and sedentary behaviour in adults with type 2 diabetes mellitus: a diary study

Louise Poppe[1], Annick L. De Paepe[2], Dimitri M.L. Van Ryckeghem[2,3,4], Delfien Van Dyck[5], Iris Maes[5] and Geert Crombez[2]

[1] Department of Public Health and Primary Care, Ghent University, Ghent, Belgium
[2] Department of Experimental Clinical and Health Psychology, Ghent University, Ghent, Belgium
[3] Section Experimental Health Psychology, Clinical Psychological Science, Maastricht University, Maastricht, The Netherlands
[4] Department of Behavioural and Cognitive Sciences, Faculty of Humanities, Education and Social Sciences, University of Luxembourg, Esch-sur-Alzette, Luxembourg
[5] Department of Movement and Sports Sciences, Ghent University, Ghent, Belgium

## ABSTRACT

**Background:** Adopting an active lifestyle is key in the management of type 2 diabetes mellitus (T2DM). Nevertheless, the majority of individuals with T2DM fails to do so. Additionally, individuals with T2DM are likely to experience mental (e.g., stress) and somatic (e.g., pain) stressors. Research investigating the link between these stressors and activity levels within this group is largely lacking. Therefore, current research aimed to investigate how daily fluctuations in mental and somatic stressors predict daily levels of physical activity (PA) and sedentary behaviour among adults with T2DM.

**Methods:** Individuals with T2DM ($N = 54$) were instructed to complete a morning diary assessing mental and somatic stressors and to wear an accelerometer for 10 consecutive days. The associations between the mental and somatic stressors and participants' levels of PA and sedentary behaviour were examined using (generalized) linear mixed effect models.

**Results:** Valid data were provided by 38 participants. We found no evidence that intra-individual increases in mental and somatic stressors detrimentally affected participants' activity levels. Similarly, levels of sedentary behaviour nor levels of PA were predicted by inter-individual differences in the mental and somatic stressors.

Corresponding author
Louise Poppe,
louise.poppe@ugent.be

## INTRODUCTION

Diabetes mellitus is a significant problem in Western society and the number of individuals with diabetes mellitus is still growing (*International Diabetes Federation, 2019*). In 2019, 8.9% of European adults lived with (un)diagnosed diabetes mellitus (*International Diabetes Federation, 2019*). The most common form of diabetes mellitus is type 2 diabetes mellitus (T2DM), accounting for 90% of all diabetes cases (*International Diabetes*

*Federation, 2019*). Adopting and maintaining an active lifestyle, with sufficient physical activity (PA) and a limited amount of sitting time, is considered key in the prevention and management of T2DM (*Dempsey et al., 2016*; *Sigal et al., 2006*). Similar to the general population, individuals with T2DM are recommended to accumulate 150–300 min of moderate-intensity PA, 75–150 min of vigorous-intensity PA, or an equivalent combination of moderate- and vigorous-intensity PA throughout the week and to do muscle-strengthening activities on two or more days per week (*WHO, 2020*). Furthermore, adults with T2DM are encouraged to minimize the amount of time spent being sedentary (*WHO, 2020*) and to interrupt prolonged sitting time with bouts of light-intensity PA (LPA) every 30 min (*Colberg et al., 2016*). Yet, despite this knowledge, the majority of patients fails to reach the predefined health guidelines regarding PA and accumulates high levels of sitting time (*Hamer et al., 2013*; *Morrato et al., 2007*).

In contrast with other preventive health behaviours (e.g., getting a flu shot), adopting and maintaining an active lifestyle requires daily effort and continued dedication. These efforts are therefore likely to be influenced by daily variations in how we feel and interact with our environment (*Dunton, 2017*). Indeed, individuals act in the context of highly variable levels of mental (e.g., negative affect) and somatic (e.g., pain or fatigue) stressors (*Kanning & Schoebi, 2016*). Understanding the impact of these stressors upon activity behaviours is particular of interest for developing interventions that provide in time support to act upon the possibly detrimental impact of mental and somatic stressors (e.g., Just-In-Time-Adaptive-Interventions (*Nahum-Shani et al., 2017*)). For example, the mobile application "OnTrack" identifies potential triggers for dietary lapses (including negative mood or fatigue) and provides users with information about potential ways to cope with the detected triggers (*Forman et al., 2019*; *Goldstein et al., 2017*). Currently, this type of innovative interventions does not exist within the domain of active lifestyle adoption and maintenance (*Hardeman et al., 2019*). A key reason for a lack of similar interventions in this domain at least partially relates to a lack of understanding how mental and somatic stressors impact upon people's daily activity levels. To support future development of interventions that are tailored to the momentary state (mental and somatic) of an individual, it is key to understand how mental and somatic stressors impact upon people's activity levels, especially within populations who are often experiencing these stressors.

Individuals with T2DM often experience mental and somatic stressors in daily life (*Fritschi & Quinn, 2010*; *Heidari et al., 2019*). In comparison with the general population, people with T2DM are more likely to experience a major depressive episode (*Darwish et al., 2018*) and to experience fatigue and pain (*Heidari et al., 2019*). As yet, no research has examined whether within-person alterations in mental and somatic stressors affect activity levels throughout the day in individuals with T2DM.

A few studies in non-patient populations have already examined how intra-individual changes in mental and somatic stressors influence people's activity levels. Dunton and colleagues demonstrated that, within adults aged 50 years and above, momentary negative affect predicted lower levels of self-reported moderate-to-vigorous PA (MVPA) in the subsequent 4 h interval after the assessment (*Dunton et al., 2010*). Similarly, Zenk and
colleagues found that negative affect measured in the morning was associated with lower levels of subsequent daily accelerometer-measured MVPA and higher levels of subsequent daily accelerometer-measured sedentary behaviour in African-American women aged 25–64 years (*Zenk et al., 2017*). Elevated levels of momentary stress and fatigue have also been found to decrease subsequent levels of accelerometer-measured PA in adults (*Jones et al., 2017*; *Liao et al., 2017*; *Vetrovsky et al., 2021*).

Yet, studies examining the impact of intra-individual fluctuations in mental and somatic stressors on activity levels within patient populations, however, are scarce. Research in people with fibromyalgia and/or chronic fatigue syndrome showed that momentary pain and fatigue predicted decreased accelerometer-measured activity levels in the 30 min interval after the assessment (*Kop et al., 2005*). Similarly, Murphy and colleagues found that increased momentary fatigue predicted reduced levels of accelerometer-assessed PA in the subsequent 4 h among people with symptomatic knee and hip osteoarthritis (*Murphy et al., 2012*).

The aim of the current study is to address the research gap regarding the daily impact of mental and somatic stressors on the activity levels of people with T2DM. More specifically, we aimed to assess the influence of intra-individual variations in mental and somatic stressors on daily levels of accelerometer-measured PA and sedentary behaviour in individuals with T2DM.

## MATERIALS & METHODS

### Participants

Data-collection for this study was part of the baseline test of a randomized controlled trial examining the efficacy of an online intervention promoting an active lifestyle (*Poppe et al., 2019b*). Consequently, the a priori power analysis was based on the targeted outcomes of the randomized controlled trial (*Poppe et al., 2019a*). Based on this power analysis we aimed to recruit 96 people with T2DM. Individuals with T2DM were recruited via different channels. First, people with T2DM were recruited via the Ghent University Hospital and the Damian General Hospital (Ostend). Because recruitment via the hospitals was slower than expected, the study was also advertised via the Flemish Diabetes Association. Finally, people with T2DM who participated in previous research (*Poppe et al., 2017*) were invited to participate in the randomized controlled trial. To be eligible, patients had to (1) be diagnosed with T2DM for at least one month, (2) be 18 years or older, (3) be Dutch-speaking, (4) be computer-literate and (5) have Internet access. The study was conducted between January and August 2018. The study was approved by the Committee of Medical Ethics of the Ghent University Hospital (Belgian registration number: B670201732566). The protocol for the trial was published (*Poppe et al., 2019a*) and all participants provided a written informed consent.

### Procedure

After enrolment, participants were visited at home by a researcher. Participants' weight and waist circumference were assessed. Next, participants were instructed to complete an ad-hoc questionnaire assessing demographic information. Finally, they were instructed to

wear an accelerometer and to fill out an online morning diary for 10 consecutive days starting the day after the home visit. After this period of 10 days, the accelerometer was recollected and participants were randomized to the intervention group or the waiting-list control group.

## Measurements

### Demographic information

Participants' age, sex, height, civil status, level of education, profession and time since diagnosis were assessed via an ad hoc questionnaire. Level of education was dichotomized in "low" (primary or secondary education) versus "high" (college or university). A Seca weighting scale (type 813) and a Seca measuring tape were used to determine participants' weight and waist circumference. To minimize measurement error, participants' weight and waist circumference were assessed twice. In case there was a difference larger than 100 g or 1 cm, the measurement was conducted a third time. The mean of the measurements was considered as the final score.

### Mental wellbeing

During the home visit, participants' levels of anxiety and depression were assessed using scales of the Patient-Reported Outcomes Measurement Information System (PROMIS) (*Cella et al., 2010*). The depression short-form scale (version 1.0) and the anxiety short-form scale (version 1.0) each contain six questions with five answer options (i.e. "never", "seldom", "sometimes", "often" and "always") and assess feelings of anxiety and depression in the past seven days.

### Activity levels

Participants' levels of PA and sedentary behaviour were assessed using ActiGraph accelerometers (type GT3X+) worn on the right hip. The validity and reliability of accelerometers are influenced by the selected cut points for data reduction. As our sample's mean age was 63 years (see Results), cut points for older adults described by Barnett et al. (*Barnett et al., 2016*) were used to categorize each minute of wear time as sedentary (0–25 counts per minute (CPM)), LPA (26–1,012 CPM) or MVPA (≥1,013 CPM). In the validation study of Barnett et al. older adults were instructed to wear the GT3X+ accelerometer and a GPS monitor on the right hip during overground walking. Participants' energy expenditure was assessed using a mobile breath-by-breath gas analysis system while they walked at different walking speeds. Similarly, *Aguilar-Farías, Brown & Peeters (2014)* found that the hip-worn ActiGraph GT3X showed good accuracy for detecting sedentary behavior in older adults in free-living environments using the <25 CPM cut point. In this study the ActiGraph GT3X was compared with an inclinometer (i.e. the ActivPal) (*Aguilar-Farías, Brown & Peeters, 2014*). Finally, *Aadland & Ylvisåker (2015)* showed that the hip-worn ActiGraph GT3X is a reliable tool for assessing intensity-specific PA and sedentary behavior in adults under free-living conditions when measurements are performed over multiple days (*Aadland & Ylvisåker, 2015*).

Participants were instructed to wear the accelerometer during waking hours and to remove the device for water-based activities (e.g., bathing). The accelerometer was

initialised and the data were processed using ActiLife 6.13.3 software (ActiGraph, Fort Walton Beach, FL, USA). An epoch was set at 60 s and periods of more than 60 min of consecutive 0 counts were considered as non-wear time (*Troiano et al., 2008*). A valid day was considered as a day with a minimum of 600 min of wear time (*Troiano et al., 2008*). For each participant the number of minutes per day spent sedentary, performing LPA and performing MVPA were calculated.

### Daily mental and somatic stressors

The morning diary was created with the survey software LimeSurvey (LimeSurvey Project, Hamburg, Germany). All participants received an e-mail with a link to the website and a unique token to log in. Participants filled out the morning diary using their personal computer or tablet and could only access and complete the morning diary between 3 AM and 11 AM. The morning diary assessed mental and somatic stressors that are prevalent in individuals with T2DM (i.e., fatigue (*Fritschi & Quinn, 2010*; *Heidari et al., 2019*), stress (*Hackett & Steptoe, 2017*; *Qiu et al., 2017*), sadness (*Ali et al., 2006*; *Darwish et al., 2018*), pain (*Heidari et al., 2019*; *Kirk et al., 2019*), nausea/dizziness (*Heidari et al., 2019*) and numbness/tingling in the limbs (*Aikens, 1998*; *Kästenbauer et al., 2004*)). All mental and somatic stressors were assessed using single-item measures. This is in line with numerous diary studies (e.g., *Bouwmans et al. (2017)*) and was done to limit the time needed to complete the diary and hence minimise the chances of drop-out or random responses. Additionally, considering the repeated assessment in daily life, traditional questionnaires assessing mental and somatic stressors are not by default valid for this type of research (*Degroote et al., 2020*). Participants were instructed to indicate how strongly they experienced the stressor "right now" using a 10-point scale, ranging from 1 "(absolutely not)" to 10 "(very much)".

To ensure comprehensibility of the items assessing the mental and somatic stressors, a cognitive interview was conducted with four volunteers (mean age = 58.3 years (SD = 6.5); 75% women; 50% with a high level of education (i.e. college/university); 50% diagnosed with T2DM) (*Beatty & Willis, 2007*; *Poppe et al., 2019a*). For these cognitive interviews, the volunteers were instructed to answer each item of the morning diary and to explain how they came to this answer. Several adaptations were performed to the initial items, assuring the unambiguous comprehension of the items assessing the mental and somatic stressors. To avoid possible interference, these volunteers did not take part in the current study.

## Data-analysis

The data were analysed using R version 3.2.5 (*R Development Core Team, 2010*). To ensure the quality of the data, participants' data were only included when they (1) filled-out the diary for a minimum of 7 days (*Rost et al., 2016*; *Van Ryckeghem et al., 2013*) and (2) had a minimum of 4 valid accelerometer days (*Van Dyck et al., 2019*).

To take into account the clustering of the data within participants (generalized) linear mixed effect models as implemented in the package lme4 version 1.1–19 (*Bates et al., 2014*) were used to analyse the data. Considering inter- and intra-individual differences in

accelerometer wear time, participants' number of daily minutes spent performing sedentary behaviour, LPA or MVPA were divided by the daily number of minutes that participants had worn the accelerometer (*Hooker et al., 2016*). Levels of within- and between-subject variance of the stressors and the activity levels were calculated by running intercept-only models (i.e. models only including a fixed and random intercept) with each of the stressors as well as participants' levels of sedentary behaviour, LPA and MVPA as outcome variable.

To examine the effect of the stressors on participants' levels of sedentary behaviour, LPA and MVPA, (generalized) linear mixed effect models were fitted with the mental and somatic stressors as between-subject (i.e., mean of the variable at the subject-level) as well as within-subject (i.e. individuals' daily score minus their mean score) variables. This was done because the standard mixed model approach does not distinguish between these within- and between-cluster effects and implicitly assumes these effects are the same. However, incorrectly assuming common effects can obscure the association of covariates with the response. By modelling the within-and between-subject effects separately, the discrepancy of these effects becomes explicit (see also *Neuhaus & Kalbfleisch (1998)*). Participants' age, sex (i.e., male vs. female), waist circumference, level of education (i.e., low vs. high), retirement status (i.e. retired vs. not retired), level of anxiety (PROMIS) and level of depression (PROMIS) were also included in the model. To facilitate convergence, the variables "age", "waist circumference", "level of anxiety (PROMIS)" and "level of depression (PROMIS)" were standardized before they were entered in the model.

For each fitted model the normality assumption was checked by visually inspecting the residuals versus fitted values plot and the quantile-quantile plot. If normality could not be assumed, the Bayesian Information Criterion (BIC) of models with different variance and link functions (i.e., Gaussian with identity, Gamma with log, Gamma with identity, Poisson with log, and negative binomial with log) were compared and the model with the lowest BIC value was selected (*Germeys & De Gieter, 2018*).

## RESULTS

### Data availability and descriptive statistics

Despite the intensive recruitment process, only 54 adults with T2DM agreed to participate in the study. After correcting for the required number of valid accelerometer and diary days, data of 39 participants was retained. Supplemental File 1 provides a side-by-side comparison of the characteristics of participants who provided valid accelerometer and diary data and those who did not. Demographic information of one participant was missing. Consequently, analyses were performed on the data of 38 participants. Table 1 provides participants' demographic characteristics.

On average, participants wore the accelerometer for 9 days (SD = 2) and completed the diary on 9 days (SD = 1). Table 2 displays participants' mean levels of PA and sedentary behaviour as well as the mean scores on the mental and somatic stressors. Five people did not reach the current PA guidelines (i.e., <150 min of MVPA per week). Supplemental File 2 illustrates the individual scores on the mental and somatic stressors over the measurement period.
| Table 1 Demographic characteristics of the participants. | |
|---|---|
| **Characteristics** | **Participants ($N$ = 38)** |
| Sex | |
| $N$ women (%) | 13 (34.21) |
| $N$ men (%) | 25 (65.79) |
| Age in years, mean (SD); range | 63.18 (7.80); 50.00–81.00 |
| Educational level | |
| $N$ high level of education (%) | 20 (52.63) |
| $N$ low level of education (%) | 18 (47.37) |
| Retirement | |
| $N$ retired (%) | 19 (50.00) |
| $N$ not retired (%) | 19 (50.00) |
| Waist circumference in cm, mean (SD); range | 109.25 (15.14); 75.55–155.75 |
| BMI (kg/m$^2$), mean (SD); range | 30.82 (6.00); 21.40–50.90 |
| Time since diagnosis in months, mean (SD); range | 129.40 (83.31); 12.00–288.00 |
| Level of anxiety, mean (SD); range | 10.18 (4.01); 6.00–24.00 |
| Level of depression, mean (SD); range | 8.63 (3.68); 6.00–23.00 |

| Table 2 Mean levels of sedentary behaviour and PA and mean scores on the mental and somatic stressors. | | | |
|---|---|---|---|
| **Variable** | **Mean (SD); range** | **Between-subject variance** | **Within-subject variance** |
| Sedentary behaviour and PA (% of wear time) | | | |
| Sitting time | 58.14 (8.18); 40.84–75.23 | 57.39 | 64.75 |
| LPA | 35.54 (7.18); 21.57–51.91 | 44.32 | 47.56 |
| MVPA | 6.31 (2.82); 0.74–13.93 | 6.14 | 15.85 |
| Stressors* | | | |
| Fatigue | 2.34 (1.31); 1.00–6.50 | 1.54 | 1.63 |
| Stress | 1.81 (1.12); 1.00–6.20 | 1.18 | 0.83 |
| Pain | 2.14 (1.44); 1.00–6.80 | 2.01 | 0.67 |
| Nausea/dizziness | 1.22 (0.57); 1.00–3.90 | 0.29 | 0.30 |
| Numbness/tingling | 1.42 (0.85); 1.00–5.33 | 0.68 | 0.45 |
| Sadness | 1.37 (0.70); 1.00–3.75 | 0.42 | 0.57 |

**Note:**
* Range of the items: 1 (absolutely not) to 10 (very much).

**Table 3 Results of the analysis for sedentary behaviour, LPA and MVPA.**

| | Sedentary behaviour | | LPA | | MVPA | |
|---|---|---|---|---|---|---|
| | **Beta** | **CI** | **Beta** | **CI** | **Beta** | **CI** |
| *Within-subject stressors* | | | | | | |
| Fatigue | 0.38 | [−0.41 to 1.26] | −0.38 | [−1.03 to 0.27] | −0.005 | [−0.06 to 0.05] |
| Stress | −0.07 | [−1.26 to 1.10] | 0.16 | [−0.77 to 1.10] | −0.04 | [−0.11 to 0.04] |
| Pain | −0.43 | [−1.72 to 0.85] | 0.55 | [−0.44 to 1.55] | 0.02 | [−0.06 to 0.11] |
| Nausea/dizziness | −1.07 | [−2.96 to 0.94] | 0.89 | [−0.82 to 2.61] | 0.02 | [−0.12 to 0.15] |
| Numbness/tingling | 0.84 | [−0.72 to 2.33] | −0.72 | [−1.80 to 0.36] | −0.04 | [−0.15 to 0.07] |
| Sadness | −0.26 | [−1.55 to 1.00] | 0.13 | [−0.99 to 1.26] | −0.01 | [−0.09 to 0.07] |
| *Between-subject stressors* | | | | | | |
| Fatigue | 1.54 | [−0.72 to 3.81] | −1.02 | [−3.80 to 1.75] | −0.03 | [−0.23 to 0.16] |
| Stress | 1.24] | [−1.62 to 4.14] | −1.34 | [−5.04 to 2.35] | −0.09 | [−0.35 to 0.17] |
| Pain | −0.40 | [−2.35 to 1.61] | 0.63 | [−1.78 to 3.03] | −0.02 | [−0.20 to 0.15] |
| Nausea/dizziness | −6.00 | [−11.25 to −0.87] | 5.57 | [−0.96 to 12.10] | 0.17 | [−0.29 to 0.63] |
| Numbness/tingling | −0.08 | [−3.24 to 3.37] | −0.49 | [−4.62 to 3.64] | 0.17 | [−0.12 to 0.46] |
| Sadness | 1.17 | [−2.44 to 4.73] | −0.19 | [−4.63 to 4.25] | −0.15 | [−0.46 to 0.17] |

**Note:**

Results controlled for age, sex, waist circumference, level of education, retirement, level of anxiety (PROMIS), and level of depression (PROMIS).

## Associations between mental and somatic stressors and activity levels

Table 3 presents the results of the analysis for sedentary behaviour, LPA and MVPA. For sedentary behaviour a Gaussian model with an identity link was used. A Gamma model with an identity link function was used to assess the effect of the stressors on participants' levels of LPA and MVPA. Intra-individual nor inter-individual differences in the stressors (i.e., fatigue, stress, pain, nausea/dizziness, numbness/tingling, and sadness) were found to be associated with alterations in participants' sedentary behaviour, LPA or MVPA ($p > 0.05$).

## DISCUSSION

This study investigated the impact of mental and somatic stressors, assessed in the morning, on accelerometer-measured PA and sedentary behaviour during that day in individuals with T2DM. Changes in the mental and somatic stressors across days were not found to be associated with changes in participants' levels of sedentary behaviour, LPA or MVPA. Similarly, levels of sedentary behaviour, LPA, and MVPA were not predicted by between-subject differences in the mental and somatic stressors.

This pattern of results was surprising. We had expected that both mental and somatic stressors would be an obstacle for participants' activity levels during that day. Individuals with T2DM may experience a variety of mental and somatic stressors, such as sadness (*Darwish et al., 2018*) and pain (*Heidari et al., 2019*), and some of these have been associated with more sedentary behaviour and low levels of PA. For example, *Chastin et al. (2014)* found that pain was one of the main drivers for being sedentary, albeit

not in a sample of patients with T2DM. There are at least three explanations for the current results.

First, we may not have detected effects because of a limited sample size. However, simulation studies have shown that a minimum of 30 clusters (in this case a minimum of 30 participants) is needed to obtain unbiased point estimates of the level 1 and level 2 fixed effects (i.e., the level of the stressor within (level 1) and between (level 2) subjects) and the fixed standard errors (*McNeish & Stapleton, 2016*). Considering that 38 participants provided valid data, we consider it unlikely that a lack of power explains our results.

Second, the mental and somatic stressors assessed in the morning may have no or a limited effect on participants' activity levels. It may well be that participants had a high maintenance self-efficacy, meaning that they were able to remain physically active in the presence of stressors (*Schwarzer, 2008*). If this is the case, it would be informative to examine how participants achieve this.

Third, the lack of effect detected might be attributed to the limited variability in the reported stressors. The majority of the participants reported not to experience the stressors "nausea/dizziness" and "numbness/tingling" when completing the diary (see Supplemental File 2). Indeed, it is possible that variability in these stressors might only be relevant for a subgroup of people with T2DM (e.g., people with neuropathy). Similarly, limited variation over days was found for the stressor "sadness" as most participants reported not to experience this stressor when completing the diary. Röcke et al. instructed 19 older adults to rate their positive and negative affect for a period of 45 days. Similarly, the mean score for negative affect was low (i.e., 1.32 on 8) and the majority of participants (i.e. 13 out of 19) provided the same rating score for "sad" on more than 90% of the 45 days (*Röcke, Li & Smith, 2009*).

More variation over days was found for the stressors "fatigue", "stress" and "pain". Nevertheless, no evidence for an association between these stressors and participants' activity levels was found. *Vetrovsky et al. (2021)* assessed older adults' morning fatigue and daily PA for a period of four weeks and found that greater morning fatigue was associated with less accelerometer-measured daily MVPA (*Vetrovsky et al., 2021*). However, in this study depressive symptoms, which might influence the stressor (i.e. fatigue) as well as the outcome (i.e. levels of sedentary behaviour and PA) were not taken into account. Furthermore, in contrast with our study, the within- and between-subject effects were not separately modelled. Similar to our results, Liao et al. found no between- or within-person association between levels of fatigue and levels of LPA and MVPA in the subsequent 30 min. time window among healthy adults (mean age = 40.4) (*Liao et al., 2017*). However, in the study of Liao et al. daily increases in negative affect (including feelings of depression and stress) did predict an increase in LPA in the subsequent 30 min. time window. Nevertheless, it should be noted that mean levels of negative affect were higher in their study (i.e., 1.44 on a 5-point scale) than in ours (i.e., 1.37 (sadness) and 1.81 (stress) on a 10-point scale). In the current study the reported levels of the mental and somatic stressors were overall low (i.e. the mean score ranged between 1.21 (nausea/dizziness) and 2.34 (fatigue) on 10). Consequently, the level of these stressors might have been too low to actually influence participants' levels of sedentary behaviour, LPA or MVPA. A first

potential explanation for these low scores within our sample is self-selection: individuals with lower levels of mental or somatic stressors might be more likely to participate in an activity-promoting programme than individuals facing higher levels of these stressors. The low scores for the stressors "fatigue", "stress" and "pain" can also be explained by the timing of the diary (i.e., in the morning). It is possible that the intensity of these stressors is relatively low when waking up, but increases throughout the day. Similarly, *Vetrovsky et al., 2021* reported low levels of morning fatigue in their study focusing on the relation between morning fatigue and PA among older adults. Further research implementing multiple measurements throughout the day is needed to elucidate these findings.

This study has several strengths. First, according to our knowledge, this is the first study examining the within-subject effects of mental and somatic stressors on physical activity and sedentary behaviour in individuals with T2DM. Second, participants' activity levels were assessed using accelerometers rather than self-report measures. In doing so the impact of recall biases was minimized. Third, completion rates of the morning diary and adherence to the accelerometer protocol were high. On average, participants filled out the diary and wore an accelerometer on 9 of the 10 days.

There are also a number of limitations. First, according to the accelerometer data, participants' activity levels were already quite high (i.e., 49 min/day of MVPA on average). However, the cut points for classifying participants' activity levels are based on research with older adults having a mean age of 70 years (range: 60–87.6 years) (*Barnett et al., 2016*). Because the mean age of our sample was 63 years (range: 50–81 years), it is possible that we overestimated participants' activity levels. When applying the Freedson cut points for adults (*Freedson, Melanson & Sirard, 1998*), our sample accumulates on average 21 min of MVPA per day and 19 participants do not meet the current PA guidelines (data not shown). Furthermore, it is important to note that our main interest was to investigate the relation between mental and somatic stressors and activity levels rather than participants' mean activity levels. Second, the mental and somatic stressors were only measured in the morning. These data were then used to predict participants' activity levels over that day. Other studies investigating shorter time frames (e.g., the impact of stressors on activity levels in the following 15 min (*Jones et al., 2017*), 30 min (*Kop et al., 2005*) or 4 h (*Dunton et al., 2010*; *Murphy et al., 2012*) after the measurement of the stressor) found more evidence for within-subject associations between mental and/or somatic stressors and people's activity levels. Indeed, our lengthy time frame (i.e. one day) might have masked short-term effects of intra-individual increases in the mental and somatic stressors. More advanced designs including ecological momentary assessment with multiple measurements throughout the day may overcome this problem. However, as this study was part of the baseline test of a randomized controlled trial, it was decided not to overwhelm participants with a more intensive design of data collection. Third, although we aimed to adjust the analysis for the most relevant confounding variables, confounding from other variables remains a possibility. For example, medication adherence was not assessed in the current study. Hence, we were unable to adjust the analysis for medication effects. Finally, the participants of this study had agreed to take part in a randomized controlled trial testing the effect of an online intervention targeting an active lifestyle and

were only included when they met the inclusion criteria. It is therefore possible that the participants differed on several domains from people who refused to participate (e.g., age or severity of diabetes-related complications). As self-selection might have occurred, one should be cautious to extrapolate our results to the overall population of adults with T2DM.

The current findings have implications for intervention development as well as for further research. In order to be relevant targets for interventions tailored to the momentary state of individuals, mental or somatic stressors should (1) vary over time and (2) predict individuals' activity levels. In line with previous research (Röcke, Li & Smith, 2009), we found limited variation over days for the stressor "sadness". Hence, this stressor might be of limited interest for the development of interventions tailored to the momentary state of people with T2DM. Higher variation over days was detected for the stressors "fatigue", "stress" and "pain". However, considering the lack of evidence found for an association between these stressors and participants' activity levels, interventions targeting these stressors might have limited impact. Two somatic stressors, namely "numbness/tingling" and "nausea/dizziness" only varied within a subgroup of our sample. Further research is needed to examine the impact of these stressors in specific subgroups of people with T2DM (e.g., people with neuropathy).

## CONCLUSIONS

The aim of this study was to investigate whether fluctuations in mental and somatic stressors across days affect daily levels of PA and sedentary behaviour in adults with T2DM. No detrimental effect of intra-individual increases in mental and somatic stressors on participants' activity levels was detected. Similarly, levels of sedentary behaviour, LPA, nor MVPA were predicted by between-subject differences in the mental and somatic stressors. This study is the first to investigate within-subject effects of mental and somatic stressors on physical activity and sedentary behaviour in individuals with T2DM. Studies adopting ecological momentary assessment with multiple measurements throughout the day are needed to gain more insight into the potential short-term effects of mental and somatic stressors on the activity levels of people with T2DM.

### Funding

This study was funded by the Fund for Scientific Research Flanders (FWO—Flanders) (11Z4716N). The funders had no role in study design, data collection and analysis, decision to publish, or preparation of the manuscript.

### Grant Disclosures

The following grant information was disclosed by the authors:
Fund for Scientific Research Flanders (FWO—Flanders): 11Z4716N.

## Competing Interests

The authors declare that they have no competing interests.

## Author Contributions

- Louise Poppe conceived and designed the experiments, performed the experiments, analyzed the data, prepared figures and/or tables, authored or reviewed drafts of the paper, and approved the final draft.
- Annick L. De Paepe conceived and designed the experiments, analyzed the data, authored or reviewed drafts of the paper, and approved the final draft.
- Dimitri M. L. Van Ryckeghem conceived and designed the experiments, performed the experiments, authored or reviewed drafts of the paper, and approved the final draft.
- Delfien Van Dyck conceived and designed the experiments, authored or reviewed drafts of the paper, and approved the final draft.
- Iris Maes conceived and designed the experiments, authored or reviewed drafts of the paper, and approved the final draft.
- Geert Crombez conceived and designed the experiments, authored or reviewed drafts of the paper, and approved the final draft.

## Human Ethics

The following information was supplied relating to ethical approvals (i.e., approving body and any reference numbers):

The study was approved by the Committee of Medical Ethics of the Ghent University Hospital (Belgian registration number: B670201732566).

## Data Availability

The R script and datasets are available in the Supplemental File.

## Supplemental Information

Supplemental information for this article can be found online at http://dx.doi.org/10.7717/peerj.11579#supplemental-information.

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
