# Peer review of "The impact of mental and somatic stressors on physical activity and sedentary behaviour in adults with type 2 diabetes mellitus: a diary study"

_PeerJ, doi:10.7717/peerj.11579_

## Round 0.1 · original submission · Major Revisions

As you will see from the reviews below, both reviewers found your work to be of interest and have agreed that it should be published. Both have raised a number of points, which if addressed I think will add considerably to the manuscript, and assist in clarity and interpretation. Hence, I am inviting you to consider and respond to the suggestions and submit a revised manuscript.

Please address each of the points in turn (or rebut any with which you disagree). Although there is a fairly long list, as you will see many of these are not major; I have indicated 'major revisions' to reflect the range of issues raised, rather than their 'difficulty' in addressing. Reviewer-1 has suggested further/alternative analysis. we would welcome a discussion on this point if you feel this isn't warranted or achievable in these difficult circumstances.

Thanks for submitting to PeerJ.

·

Basic reporting

- overall clear, concise, and well written
- line 44 - I'd suggest using the current prevalence of diabetes in Europe rather than projections, they are already stark enough
- line 50 - the WHO reference is the old guidelines, check the new 2020 WHO guidelines - I believe they may have a section on populations with chronic conditions too
- line 74 - could you clarify by what you mean by people T2DM more likely to have a low mood? As in a mood disorder?
- line 256 - I'd drop the first two sentences of the discussion, you only need to state the aims of the current study here then introduce your main findings right away
- in describing the results from each model in the results and discussion, it would be useful to include a clear metric of effect size as well as stating the statistical significance. E.g., higher level of stress was associated with 10 fewer minutes of LPA per day compared to low.
- clearly described objectives and encouraging to see power analysis to support
- line 139 - the accelerometers were shown to be reliable and valid, how? Good to give detail about what you mean by reliable and valid, e.g., is it validated against doubly labelled water etc.? Crucially, is it validated for use in people with T2DM? Do we know the results of the reference validation studies would be generalisable to people with T2DM? Another more technical point is that they can't assess 'sitting', which requires an inclinometer to determine if someone is seated or standing. They can approximate sedentary behaviour - although though many would question its validity of doing so given the validation studies are mostly for MVPA or total physical activity volume. I'd suggest referring only to sedentary behaviour throughout, rather than sitting
- Again, important to clarify when participants wore the accelerometer because your estimation of sedentary behaviour (0-25 CPM) would include sleep. If participants did wear it 24h, you'd have subtracted sleep time I assume?
- line 152 - clarify if the survey completed on a computer? Or tablet/phone? Also, state here what times participants completed the surveys in the morning i.e. it would be helpful to have an idea of where they were/what they were doing at the time, if possible
- line 195 - include a sentence on why you've standardised the age and waist circumference variables here

Experimental design

- line 139 - the accelerometers were shown to be reliable and valid, how? Good to give detail about what you mean by reliable and valid, e.g., is it validated against doubly labelled water etc.? Crucially, is it validated for use in people with T2DM? Do we know the results of the references validation studies would be generalisable to people with T2DM? Another more technical point is that they can't assess 'sitting', which requires an inclinometer to determine if someone is seated or standing. They can approximate sedentary behaviour - although though many would question its validity of doing so given the validation studies are mostly for MVPA or total physical activity volume. I'd suggest referring only to sedentary behaviour throughout, rather than sitting
- Again, important to clarify when participants wore the accelerometer because your estimation of sedentary behaviour (0-25 CPM) would include sleep. If participants did wear it 24h, you'd have subtracted sleep time I assume?
- line 152 - clarify if the survey completed on a computer? Or tablet/phone? Also, state here what times participants completed the surveys in the morning i.e. it would be helpful to have an idea of where they were/what they were doing at the time, if possible
- line 195 - include a sentence on why you've standardised the age and waist circumference variables here This may a little out of scope, but did you consider some form of trajectory modelling for this analysis in addition to the main analysis? It seems to me quite a bit of information loss occurs when using the mean daily scores of mental/somatic stressors - these are dynamic processes that will vary day-to-day. It would be interesting to see if a trajectory model identifies latent subgroups, such as people with highly varying stress, who have different activity patterns to those with a constant experience of high or low stress
- line 196 - the type of day was 'entered into the model' presumably as a confounding variable? Good to clarify this. Also, I don't quite see why it would be a confounding variable in this study. This would assume it is a common cause of both the exposure (mental/somatic stressors) and the outcome (activity). Is there evidence that type of day affects mental/somatic stressors as well as activity? If so, state this too as it isn't clear to me
- Otherwise nice reporting of the statistical analysis, this is a downfall of many papers I come across
- line 211 - great to see consideration of sufficient power for the analysis from simulation studies
- Do you have demographic information on who declined to participate and who didn't have enough accelerometer data? It would be useful to have a side-by-side comparison with those included in the study (possibly in a supplementary) to assess selection/attrition bias
- tables 3-5 - unless there is a good reason for doing so, I would avoid reporting all the confounding variable coefficients to minimise the risk of Table 2 fallacy. Either move these full tables into the supplementary or remove completely. I'd then combine these tables into one - there isn't really a need to present these models separately, it breaks up the paper unnecessarily
- line 256 - in discussing main findings, also report the null findings, these are interesting too e.g., no association with MVPA

Validity of the findings

- it would be helpful to see a clearer interpretation of the implications here - what do these findings really mean? I'm struggling to make sense of them myself, what do they mean for interventions to promote activity in people T2DM? There were a lot of null findings that could also be of interest - was stress not related to MVPA? Is stress not a factor in the difficulties promoting MVPA then?

- mental health was not adjusted for, which is surprising to me, particularly depression and anxiety symptoms which can explain both the exposure and the outcome. Is the data available? It's hard not to think these findings are highly confounded by existing mental health symptoms

-given time is finite throughout the day, it is strange to find associations between stress and only one activity domain e.g., light but not sedentary or MVPA. The increase in light activity must be coming at the expense of something else, can the authors explain this discrepency? Perhaps a more sophisticated analysis would be more suitable here. Compositional data analysis would handle the time-dependent nature of each activity variables more appropriately

Additional comments

Overall, an interesting study on an interesting - and overlooked - topic. The paper describes the methods and rationale well, with clear and concise language. The analysis is also well described.

Reviewer 2 ·

Basic reporting

Poppe et al have examined the impact of mental and somatic stressors on physical activity and sitting time in 39 participants with T2D. Participants were asked to wear an accelerometer for 10 days and to fill out a morning diary assessing mental and somatic stressors on each day. Overall, the manuscript is well-written and would make a useful contribution to the literature. I have a few comments, some of which are minor, that limit the interpretation of the findings.

Experimental design

* More information regarding the participants would be helpful. For example, how prevalent were pre-existing co-morbidiites/complications that could have influenced the results?

* Similarly, do you have any information regarding medications? Insulin use is probably important as well as other glucose/cholesterol lowering medications that potentially have side effects related to outcomes

Validity of the findings

* You lose a reasonable amount of people with your restrictions. Do the results change if you include those with only 1 day of accelerometer data? If not, this would allow you to increase your sample size. Limiting to 4 days could always be included as a sensitivity analysis

* Although mean MVPA was high, there appears to be a reasonable degree of variance. I think it would be useful to the reader to report how many of your participants actually reached the current PA recommendations. This may subsequently require some rewording in the introduction to say that adopting or maintaining an active lifestyle is important in the management of T2D.

* It should be mentioned in the limitations that despite adjusting for some potential cofounders, residual confounding or confounding from unmeasured factors remains a possibility

Additional comments

Abstract

* This should stipulate that the analysis included 39 participants. Also, the between subject effect of stress on sitting time should be included. If possible, the results of the linear models should also be included.

Introduction

* Latest WHO guidelines have recently been published. They make specific reference to those with chronic disease and should be included in the introduction.

* It is important to include the Colberg (2016) ADA guidelines in the introduction. They make more specific recommendations regarding sitting time.

* Although the introduction is well written, it incudes superfluous information. In my opinion, removing lines 89-97 would help to streamline

Methods

* There is a need to include accelerometer wear time as a covariate, or present as the % time spent in each behaviour during waking hours.

* Also, I appreciate your numbers are small and adjusting for too many covariates reduces the degrees of freedom, but what is the rationale for not including time since diagnosis as a covariate?

Discussion

* Line 294 - This seems a bit too speculative. It is more common in T1 and you have no data to suggest any of your participants suffer from gastroparesis.

* Lines 342-346 could be removed, as this text appears again in the conclusions

Minor comments

* Line 105 - I assume they were randomised after wearing the accelerometer?

* Please can you include the accelerometer model

---

## Round 0.2 · accepted · Accept

Thank you for the detailed response to the suggestions of the reviewers. I am happy to now accept this work - congratulations.

Reviewer 2 ·

Basic reporting

No comment

Experimental design

No comment

Validity of the findings

No comment

Additional comments

The authors have done a good job in addressing the previous comments. Although the findings/interpretations have changed, I feel the scientific rigour of the manuscript has improved